# Effect of Modified and Unmodified Oak Bark (Quercus Cortex) on the Cross-Linking Process and Mechanical, Anti-Aging, and Hydrophobic Properties of Biocomposites Produced from Natural Rubber (NR)

**DOI:** 10.3390/ma17091968

**Published:** 2024-04-24

**Authors:** Aleksandra Smejda-Krzewicka, Konrad Mrozowski, Krzysztof Strzelec

**Affiliations:** Institute of Polymer and Dye Technology, Faculty of Chemistry, Lodz University of Technology, Stefanowskiego Street 16, 90-537 Lodz, Poland; krzysztof.strzelec@p.lodz.pl

**Keywords:** hydrophobicity, biocomposites, oak bark, bio-filler, contact angle, natural rubber, anti-aging properties, wood waste

## Abstract

The study explores the novel use of oak bark (Quercus cortex) as a bio-filler in elastomeric composites, aligning with the global trend of plant-based biocomposites. Both modified and unmodified oak bark were investigated for their impact on the physicochemical properties of natural rubber (NR) composites. The bio-filler modified with n-octadecyltrimethoxysilane exhibited enhanced dispersion and reduced aggregates in the elastomeric matrix. NR composites containing more than 20 phr of unmodified and modified oak bark demonstrated an increased degree of cross-linking (α_c_ > 0.21). Mechanical properties were optimal at 10–15 phr of oak bark and the sample with modified bio-filler (10 phr) achieved the highest tensile strength (15.8 MPa). Silanization and the addition of the bio-filler increased the hardness of vulcanizates. The incorporation of oak bark improved aging resistance at least two-fold due to phenolic derivatives with antioxidant properties. Hydrophobicity decreased with added bark, but silanization reversed the trend, making samples with a high content of oak bark the most hydrophobic (contact angle: 129°). Overall, oak bark shows promise as an eco-friendly, anti-aging filler in elastomeric composites, with modification enhancing compatibility and hydrophobicity.

## 1. Introduction

Over the past decade, biocomposites have gained a lot of popularity in the field of materials science. They include bio-fillers derived from natural sources, which are promising candidates for enhancing the properties of polymers [1,2,3]. These offer a sustainable approach to enhance material performance while minimizing the carbon footprint [4,5,6]. However, natural fillers must meet certain requirements to be used as fillers, such as chemical neutrality, affinity to the polymer, and stability during processing and use [7,8]. In the rubber industry, natural fillers may act as reinforcing fillers. Natural fillers include flax, hemp, fibers (such as coconut), oak bark (Quercus cortex), nut, or rice husks. The advantage of organic fillers is their low price, availability, and renewability. In addition to the durability aspects, additional advantages of composites with natural fillers are biodegradability and reduced CO_2_ emissions [3,4,9,10,11,12,13,14,15,16,17,18]. Moreover, bio-fillers fit in with the Sustainable Development Goals [19,20,21,22,23]. These Goals serve as a roadmap for creating a more sustainable and improved future for everyone. They tackle worldwide challenges such as poverty, inequality, climate change, environmental damage, peace, and justice. According to a study [19], the use of biocomposites, compared with conventional materials, lowers the carbon footprint of the product. According to a life-cycle analysis of the product, the use of natural fillers in composites compared with traditional ones contributes to a decrease in CO_2_ emissions of about 20–50%. Plant-based materials are ideally suited to the 12th and 13th Sustainable Development Goals, aimed at responsible production and consumption and tackling global climate change [21,22,23].

One of the most popular bio-fillers is cellulose, which is obtained from trees, leaves, and grasses [24,25]. Its fibers can be used as rubber modifiers to improve elasticity and mechanical properties. However, adding cellulose above 10 phr may decrease elasticity and slightly lower tensile strength. The optimum amount of cellulose in the elastomeric matrix is in the range of 10 phr to 15 phr [24,26,27,28]. Nonetheless, other studies have shown that adding cellulose to a polymeric matrix results in an increase in thermal stability, crystallization temperature, and mechanical strength [24,25,26,27,28,29,30,31,32]. Adding cellulose to polyurethane foams results in materials that retain water for a long period [33].

In the search for novel solutions, few researchers have focused on applying tree bark as a filler in polymeric matrixes [13]. One reason for using this material as an additive is the significant amount of waste generated during the wood production process, such as chips, sawdust, and bark [34]. According to the literature, it has been estimated that the annual global amount of bark produced is about 359,111,200 m^3^ [35]. A considerable portion of this raw material has been used in landscaping or it has been used as an energy resource. The huge potential of bark in many applications is due to its high content of various organic compounds such as tannins, mainly ellagitannins, and gallotannins. These compounds function as antioxidants, dyes, natural fungicides, and insecticides; hence, it has important anti-aging properties [36,37]. Moreover, bark is also a source of catechin, gallocatechin, flavonoids, and proanthocyanidins [38,39,40,41,42]. For this reason, it has a wide range of applications in the polymer industry.

One of the researched applications of tree bark in materials sciences is its use as a bio-filler in UF adhesives, which are commonly used in plywood production [43,44]. Adding birch bark at 10% and 15% by weight resulted in higher shear strength and reduced formaldehyde emissions. Similar effects were observed with beech, maple, and pine bark [5,6,44,45,46]. Incorporating bio-fillers into polymers increases the hydrophilic nature of the surface, causing faster degradation of the material [47,48]. The proportion of bio-filler to the polymeric matrix needs to be chosen carefully to avoid these issues. It may be necessary to add a substance that modifies or couples the filler to the matrix for better strength parameters, proper dispersion, and enhanced hydrophobicity [48,49,50,51,52,53]. The use of bark as a bio-filler may gain importance, but different types of trees and various climates can affect the properties of composites. Therefore, it is important to identify the species of the tree from which the bark is sourced. Mirski et. al. [13] have focused on the applications of oak bark due to its abundance as a waste product from oak wood processing. Oak bark contains lignin, cellulose, holocellulose, and extractive substances, which are phenolic substances responsible for anti-aging properties [50]. Recent studies have shown that oak bark is a useful filler for melamine–urea–formaldehyde (MUF) adhesives. It not only reduces emissions during the cross-linking of MUF glue but also accelerates resin gel time. Additionally, oak bark has the potential to become a formaldehyde bio-sorbent [10,13,54,55]. The use of oak bark as a filler in elastomers is novel. Most of the related publications are based on the incorporation of cellulose or cellulose and lignin into the elastomeric matrix, not the raw bark of a tree. Investigations have confirmed the positive effect on the mechanical properties of composites with cellulose [26,27,28,29,30,31].

In addition to its applications in the field of materials science, oak bark, due to the previously mentioned active substances like tannins, flavonoids, and phenolic acids, is used in medicine, therapeutics, and health prevention. In the case of oak bark as an antiaging substance, a crucial impact relies on the content of tannins in the bark [56,57]. Their content depends on a variety of factors, such as the age of the tree, tree species, location, and section and area of the tree [58,59]. Moreover, the bark extract exhibits anti-inflammatory, antibacterial, antioxidant, and antiseptic properties [60]. Figure 1 shows the types and structures of tannins which are contained in tree bark.

This study aimed to investigate and present the characteristics of biocomposite products containing natural rubber and oak bark as a bio-filler in different amounts. Moreover, the effect of modification of oak bark with n-octadecylotrimethoxysilane was also studied. In this study, natural rubber was selected as the primary material due to its plant-based origin. It is obtained from the aqueous colloidal dispersions of rubber-bearing plants cultivated on plantations in tropical climates. To create a fully natural material, oak bark was used instead of traditional fillers such as carbon black, silica, or chalk. Oak bark is usually treated as waste during wood processing in sawmills. However, this study aimed to investigate the impact of oak bark on the properties of the elastomeric composite, making it an innovative solution for utilization of this product.

## 2. Materials and Methods

### 2.1. Materials

In this work, natural rubber (NR, RSS I) from Torimex-Chemicals Ltd. Sp. z o. o. (Konstantynów Łódzki, Poland) was used as the elastomeric matrix. To cure the elastomer, the following components were used:-sulfur (S_8_) as a cross-linking agent with a density of 1.8–2.36 g/cm^3^, supplied by Chempur (Piekary Śląskie, Poland),-zinc oxide (ZnO) as a cross-linking activator with a density of 5.6 g/cm^3^, obtained from Chempur (Piekary Śląskie, Poland),-2-mercaptobenzothiazole (MBT) as a cross-linking accelerator with a density of 1.29 g/cm^3^, delivered from Sigma-Aldrich (St. Louis, MO, USA),-stearic acid (SA), functioning as a cross-linking activator and dispersing agent with a density of 0.94 g/cm^3^, obtained from Chempur (Piekary Śląskie, Poland).

Oak bark (Quercus cortex) was used as a bio-filler, sourced from HerbaNordPol Ltd. Sp. z o. o. (Nowy Staw, Poland). For modification of the bio-filler, n-octadecyltrimethoxysilane with a density of 0.88 g/cm^3^ was employed, supplied by abcr GmbH (Karlsruhe, Germany).

As solvents to study the degree of cross-linking, the following substances were used:-toluene with a density of 0.87 g/cm^3^, delivered by POCh S.A. (Gliwice, Poland),-diethyl ether with a density of 0.71 g/cm^3^, obtained from Chempur (Piekary Śląskie, Poland).

### 2.2. Compounding and Vulcanization

The compositions of the NR mixes are outlined in Table 1. Initially, a premixture containing the requisite quantities of natural rubber, stearic acid, and zinc oxide was prepared using a two-roll mill (model: Laborwalzwerk, Krupp-Gruson, Magdeburg-Buckau, Germany) with dimensions of 200 mm in diameter and 450 mm in length, operating at a roll temperature of 30–35 °C. The total duration for creating the premix was around 10 min. Subsequently, the obtained premixtures were divided into 11 uniform portions, and the mixes were formulated utilizing the same two-roll mill setup, under identical temperature conditions. The sequence of ingredient incorporation was as follows: filler, accelerator, sulfur. Samples containing silane were prepared by initially mixing in situ the filler and silane, followed by their incorporation into the elastomeric matrix. The resulting rubber composites were individually stored in tightly sealed foils at room temperature. Vulcanization of the produced composites took place in hydraulic presses within suitable metal molds. The vulcanization parameters included a temperature of 160 °C, a pressure range of 150–180 bar, and a curing duration of 4 min.

### 2.3. Assessment of the Cross-Linking Process

The degree of cross-linking was evaluated via analyzing the cure kinetics and equilibrium swelling. The cure kinetics of the NR composites were assessed using an Alpha Technologies (Bellingham, WA, USA) oscillating disk rheometer (MDR 2000) at 160 °C, following the ASTM D5289-19a standard [61]. Various parameters were determined, including scorch time (t_02_), vulcanization time (t_90_), minimal torque (T_min_), maximal torque (T_max_), torque after 20 min of heating (T_20_), reversion losses of rheometric torque (ΔT_R_), calculated as the difference between maximal torque and torque after 20 min of heating, described by Formula (1), and cure rate index (CRI) as per the defined Formula (2):(1)$${∆T}_{R}=T_{max}-T_{20}$$
(2)$$CRI=\frac{100}{t_{90}-t_{02}}$$

Swelling characteristics were evaluated using toluene, following the ASTM D 471 standard [62]. Four test specimens, ranging from 25 to 60 mg and of various shapes, were cut from each vulcanizate, weighed using an electronic balance, and soaked in toluene until reaching equilibrium (for 72 h). Subsequently, the swollen samples were taken out from toluene, rinsed with diethyl ether, and reweighed. The samples were then dried at a constant temperature of 50 °C until reaching a stable weight and then their mass was recorded. Equilibrium volume swelling (Q_v_) was calculated using Formula (3):(3)$$Q_{v}=Q_{w}\times \frac{d_{v}}{d_{s}}$$
where Q_w_ is the value of the equilibrium mass swelling (mg/mg) calculated from Formula (4), d_v_ is the vulcanizate density (g/mL), and d_s_ is the solvent density (g/mL).
(4)$$Q_{w}=\frac{m_{s}-m_{d}}{m_{d}^{*}}$$
where m_s_ is the swollen sample weight (mg), m_d_ is the dry sample weight (mg), and $m_{d}^{*}$ is the reduced sample weight calculated from Formula (6).

The content of eluted fraction in toluene (−Q_w_) was calculated by Formula (5):(5)$${-Q}_{w}=\frac{m_{0}-m_{d}^{*}}{m_{0}}$$
where m_0_ is the initial sample weight (mg) and $m_{d}^{*}$ is the reduced sample weight calculated from Formula (6):(6)$$m_{d}^{*}=m_{d}-m_{0}\times \frac{m_{m}}{m_{t}}$$
where m_0_ is the initial sample weight (mg), m_m_ is the content of mineral substances in the compound (mg), and m_t_ is the total weight of the compound (mg).

The volume fraction of rubber in the swollen material in toluene (V_R_) was determined according to Formula (7):(7)$$V_{R}=\frac{1}{{1+Q}_{v}}$$
where Q_v_ is the equilibrium volume swelling (mL/mL).

The degree of cross-linking (α_c_) was determined using Formula (8):(8)$$\alpha_{c}=\frac{1}{Q_{v}}$$
where Q_v_ is the equilibrium volume swelling (mL/mL).

### 2.4. Determination of Surface Morphology

The examination of the vulcanizate morphology was conducted utilizing an inverted scanning electron microscope (SEM), namely, a Hitachi Tabletop Microscope TM-1000 originating from Tokyo, Japan. Sample preparation involved joining the testing sample onto a special table using double-sided self-adhesive foil. A layer of gold was then deposited onto the sample using the Cressington Sputter Coater 108 auto vacuum sputtering machine in Redding, CA, USA, under pressure exceeding 40 mbar for 60 s. Next, the prepared samples were incorporated into the scanning electron microscope chamber for analysis.

### 2.5. Determination of Mechanical Properties

For the vulcanizates, the following properties were determined: strength properties, hysteresis losses, Mullins effect, and tear resistance.

Tensile strength was evaluated using a testing machine (Zwick1435/Roell GmbH & Co. KG, Ulm, Germany) [63]. Parameters such as stress at elongation of 100%, 200%, and 300% (Se_100_, Se_200_, Se_300_), tensile strength (TS_b_), and relative elongation at break (E_b_) were determined from that test. Five samples of each vulcanizate were measured, and the test was conducted at a constant speed of 500 mm/min. The hysteresis losses were determined using a testing machine (Zwick1435/Roell GmbH & Co. KG, Ulm, Germany). Each test was conducted on three samples, which were stretched five times to 200% elongation at a stretching speed of 500 mm/min, and the initial force was 0.1 N. The Mullins effect was determined according to Formula (9):(9)$$E_{M}=\frac{W_{1}-W_{5}}{W_{1}}\times 100\%$$
where W_1_ is the hysteresis loss at the first extension of the sample (N∙mm) and W_5_ is the hysteresis loss at the fifth extension of the sample (N∙mm).

The tear strength (T_s_) was assessed according to method A outlined in ISO 34-1:2015 [64], utilizing the same testing machine used for hysteresis and tensile strength determination. Rectangular specimens, sized at 100 mm × 15 mm with a 40 mm cut, were employed for the tests.

The hardness (HA) was measured with a ZwickRoell (Ulm, Germany) hardness tester at ISO 48-4:2018 standard [65]. The samples for this test were prepared in the shape of cylinders in a specially prepared form. The measurement results were determined on the Shore A scale.

### 2.6. Resistance to Thermo-Oxidative Aging

The vulcanizates filled with oak bark underwent thermo-oxidating aging in a forced circulating aging oven maintained at 70 °C for 7 days. Following a conditioning period of 24 h at room temperature, alterations in mechanical properties, including stress at 100%, 200%, and 300% strain, tensile strength, and elongation at break, were assessed using the aging factor (AF) according to Formula (10):(10)$$AF=\frac{TS_{b}^{\prime }\cdot E_{b}^{\prime }}{TS_{b}\cdot E_{b}}$$
where $TS_{b}^{\prime }$ is the tensile strength after thermo-oxidative aging (MPa), TS_b_ is the tensile strength before thermo-oxidative aging (MPa), $E_{b}^{\prime }$ is the elongation at break after thermo-oxidative aging (%), and E_b_ is the elongation at break before thermo-oxidative aging (%).

### 2.7. Determination of Hydrophobicity

The contact angle of the vulcanizate surface was assessed utilizing a goniometer, specifically, the DataPhysics Instruments GmbH OCA 15EC from Filderstadt, Germany, employing the embedded drop method. Initially, a drop of water approximately 5 µL in volume was carefully positioned onto the surface of the vulcanizate using a Hamilton microsyringe. Subsequently, within 10 s, a photograph of the drop was captured using a specialized program, ensuring visibility of the boundary between the surface and the drop. The contact angle was then determined through analysis conducted in SCA 20 software (SCA 20 DataPhysics, Germany). A minimum of 5 drops were deposited onto each sample, and the average contact angle value was calculated.

### 2.8. Spectra Making with the FT-IR Method

The infra-red spectra of the NR vulcanizates were developed with a Thermo Scientific Nicolet 6700 FT-IR spectrometer equipped with a Smart Orbit ATR (Waltham, MA, USA) diamond attachment, using the attenuated total reflectance (ATR) method. The spectra were assessed for the wavenumber range of 3500–600 cm^−1^. Before the spectra of the samples were collected, background measurements were carried out, each time including 64 scans. Identification of the absorbance bandwidth intensities helped determine the characteristic functional groups present in the structures of the tested NR vulcanizates.

### 2.9. Statistical Analysis

To compare the differences between the tested vulcanizates, one-way analysis of variance (ANOVA) was used with the hypothesis that all tested samples were not statistically different from each other. Snedecor’s F-distribution of the results of the tested samples was also assumed and a confidence interval of 0.95 was used. Moreover, the verified null hypothesis can be presented as follows: H_0_: μ = μ_0_ against the alternatives H_1_: μ ≠ μ_0_, or H_1_: μ > μ_0_, or H_1_: μ < μ_0_. The *p*-value was determined in the program because the analysis was compared with the established significance level (α): if *p* ≤ α—reject H_0_ and accept H_1_, but if *p* > α—there are no grounds for rejecting. During the analysis, Levene’s test for homogeneity of variance test also applied to test the equality of variances. Then, to determine which specific groups differed from each other, post hoc tests were used. When the variances were equal, it was the Tukey test, and when they were not equal, it was the Games–Howell test. Statistics were compiled using the free statistical program Jamovi (version 2.5, Sydney, Australia).

## 3. Research Results and Their Discussion

### 3.1. Analysis of the Morphology of Oak Bark and NR Vulcanizates Filled with Oak Bark

The purpose of using scanning electron microscopy (SEM) was to analyze the surfaces of vulcanizates, their internal structure, and the impact of filler modification on their dispersion in natural rubber. The physical and chemical properties of rubber largely depend on the degree of dispersion and aggregation of filler particles. Therefore, understanding the composite morphology is crucial [66,67].

In Figure 2, which shows a photo of oak bark at magnification 250×, the filler particles have the shape of plates, and a lot of empty spaces between them can also be observed. Oak bark tends to form agglomerates and aggregates, as suggested by Figure 2. In addition, it can be observed that the bark is a porous material, which positively affects the possible modifications, but unfortunately, also the increased hygroscopicity of this filler.

The study of surface morphology was conducted on two samples, NR15 with 15 phr of oak bark and NR15S with 15 phr of oak bark and 1.5 phr of silane. These compositions were chosen to investigate the effect of n-octadecyltrimethoxysilane on the filler and morphology of the sample. Scanning electron microscope images were taken and Figure 3a,b represent surfaces magnified 3000 times. Figure 3a shows that the NR15 sample has larger filler aggregates than that in Figure 3b. Oak bark without modification forms aggregates and agglomerates in an elastomeric matrix due to its chemical structure and incompatibility with natural rubber. Figure 3b shows the surface of NR15S magnified 3000 times, and no such large filler agglomerates were observed. Individual small aggregates and good filler dispersion in the elastomeric matrix were noted. Hence, the filler–polymer interaction was greater than in the case of vulcanizate without added silane. Figure 4a,b show cross-sections of the samples at a magnification of 500×. It was noted that the NR15 vulcanizate (Figure 4a) had much larger oak bark particles in its structure, and they were located heterogeneously in aggregates. Moreover, particles of oak bark in the shape of plates forming large concentrations of filler in the elastomeric matrix were distinctly noticed. The incorporation of silane into the NR composite resulted in better dispersion and smaller oak bark particles in the polymeric matrix, which can be noted in Figure 4b for sample NR15S. Proper dispersion of the oak bark in the elastomeric matrix is essential for the correct effectiveness of the reinforcing action of the filler. This is achieved in this case by adding silane to the NR compositions.

### 3.2. Analysis of the Structure of NR Vulcanizates Filled with Oak Bark

The structure of NR vulcanizates before and after filling with unmodified or modified oak bark was confirmed by the analysis of infrared spectra. Figure 5 shows the spectra of three vulcanizates made of unfilled natural rubber (NR0), natural rubber filled with 20 phr of oak bark (NR20), and natural rubber filled with 20 phr of oak bark modified with 2 phr of n-octadecyltrimethoxysilane (N-OTMS). A detailed interpretation of the absorbance bands corresponding to the vibrations of characteristic functional groups occurring in the examined vulcanizate structures is presented in Table 2.

Each spectrum revealed three intensive absorption peaks at 2958, 2916, and 2848 cm^−1^ due to the stretching (symmetric and asymmetric) vibrations of -C-H in methyl (-CH_3_) and methylene (-CH_2_) groups in NR chains. The most characteristic bands for vulcanizates made of natural rubber observed on the spectrum of each tested sample were recorded in the wavenumber range of 1680–1538 cm^−1^. These peaks indicate the presence of double bonds (>C=C<) in the NR macromolecules. All spectra showed pronounced peaks at 1444, 1375, 1080, and 840 cm^−1^, which are attributed to the asymmetric scissoring vibrations of methyl groups, the symmetric scissoring vibrations of methyl groups, the wagging vibrations of methylene groups, and vibrations of residual >C=C< bonds, respectively (Table 2) [68,69]. It should be emphasized that those peaks remained unchanged during the cross-linking process for all investigated samples. The intensity of those bands may also change. The most characteristic band for this analysis, observed on the spectrum of the NR20S vulcanizate, was recorded for the wavenumber range of 1020–1005 cm^−1^. This intensive and sharp band corresponds to asymmetric stretching vibrations of the Si-O group, which confirms the binding of silane to oak bark. This peak was only visible on the spectrum of NR composite filled with oak bark modified with n-octadecyltrimethoxysilane, like a low-intensity band of deformation vibrations at the wavenumber of 1275 cm^−1^, which is typical of the Si-CH_3_ group [70]. The strong band also confirms the presence of C-Si-C groups at the wavenumber of 765 cm^−1^ and a medium-intensity doublet in the wavenumber range of 700–660 cm^−1^. Additionally, the spectrum of the NR vulcanizate filled with modified oak bark showed bands in the wavenumber range of 1454–1428 cm^−1^, which indicates the presence of methyl groups bound to silicon atoms (Si-CH_3_) [71]. These bands, observed only in the NR20S spectrum, undeniably indicate the binding of silane to the NR macromolecule, according to the diagram shown in Section 3.4.

### 3.3. The Influence of Oak Bark on the Course of NR Cross-Linking

Determination of the parameters and course of cross-linking of rubber compounds is crucial for the creation of final products. Cross-linking of rubber is related to the phenomenon of heat transfer, but also to chemical reactions that occur in the elastomeric medium. During conventional vulcanization, sulfur creates cross-bonds with the polymer chains, which cause irreversible changes in the properties of the material [67,72,73]. Therefore, the study of cross-linking kinetics yields important parameters such as the optimal vulcanization time (t_90_), scorch time (t_02_), rheometric torque increment (ΔT), and minimal rheometric torque (T_min_). These properties define the degree of cross-linking, the viscosity of the mix, the period of safe processing, and the time needed to achieve optimal properties of the final product.

Table 3 shows the vulcanization time and scorch time, as well as the minimum and maximum values of rheometric torque (respectively T_min_, T_max_) of the NR compositions tested. In addition, the reversion losses of rheometric torque are summarized to determine whether there is a reversion phenomenon. The course of cross-linking of the tested NR mixes is shown by the vulcanization curves in Figure 6. It can be seen that each composition recorded a decrease in torque after 6 min.

The study showed that the addition of oak bark did not significantly affect vulcanization and scorch times. The shortest vulcanization time (2.55 min) among the filled compositions was achieved by the NR10 sample; however, each of the tested compositions cured for between 2.5 and 3 min. The addition of oak bark caused the compositions to reach various T_min_ values. For some, this value was lower than for the reference sample, while for others, it was higher. No trend was observed to indicate whether the addition of more filler caused an increase in the minimum rheometric torque. Modification of the filler with silane resulted in an increase in T_min_ values for all compounds except for NR15S and NR20S. This phenomenon is related to better dispersion of oak bark in the elastomeric matrix and improved polymer–filler interactions. The values of the maximum rheometric torque for mixes with 20 and 25 phr of filler were greater than for other samples (4.31 dNm, 4.41 dNm, respectively), a phenomenon that may also be related to the formation of agglomerates and aggregates in the elastomeric matrix, which increased the viscosity of the composition and caused the growth of its rheometric torque values. Samples with silane achieved lower T_max_ values than those without silane (for example, NR10—3.98 dNm; NR10S—3.74 dNm). The NR20S and NR25S samples received higher T_max_ values than other compounds with silane addition (respectively, 3.81 dNm and 3.95 dNm). Measured rheometric torque increment after 20 min indicated that each of the tested rheometric curves reversed during curing, so it is important to conduct this for the most optimal time. The lowest T_20_ was achieved by the NR5S sample (2.94 dNm), but this did not indicate the greatest reversion on the rheometric curve, because the parameter ΔT_R_ specifies how great is the reversion. The higher the ΔT_R_ value, the more the reversion, causing the degradation of sulfide bonds and the deterioration of mechanical properties. If the elastomeric mix is heated for too long, it becomes over-cross-linked and loses its unique physical and chemical properties irretrievably. Among the compounds tested, the NR25 sample recorded the smallest losses on reversion (0.51 dNm). Overall, the addition of oak bark resulted in a reduction of reversion losses of rheometric torque relative to the reference sample (Figure 6). Regarding the CRI indicator, it was noteworthy that the NR5S composition achieved the highest value (86.21 min^−1^). The addition of a larger amount of oak bark caused a decrease in CRI values in most cases (except NR10), suggesting that the larger the amount of bark, the longer the curing process. It was important to note that the scorch time for all samples was approximately 1.5 min. This was particularly essential when processing these samples, as a short scorch time can lead to issues during the cross-linking process. When the mix loses its plasticity, it can become difficult to work with.

For further analysis of the degree of cross-linking of NR vulcanizates, equilibrium swelling tests were carried out to determine the resistance of the obtained samples to solvents (in this case, toluene). The results in Table 4 show that the incorporation of above 20 phr of oak bark (4.48 mL/mL) caused a decrease in the equilibrium volume swelling value (Q_v_). For samples with silane, this trend was not observed, and for these samples, the Qv value changed regardless of the quantity of filler. The content of eluted fraction (−Q_w_) for all samples reached similar values, and it was found that the NR composite was resistant to toluene, as the value was about 0.06 mg/mg. The volume fraction of rubber in the swollen sample (V_R_) showed a similar dependence to that described for Q_v_. The presence of filler increased the V_R_ value, and silane modification had an inconsistent effect on this value compared with vulcanizates with bark addition only. The incorporation of filler in the NR20 and NR25 samples resulted in the highest α_c_ values of 0.223 and 0.234, respectively. This indicates that the oak bark increased the degree of cross-linking in rubber products. However, the addition of silane did not improve the degree of cross-linking in most cases. Samples with 10 and 15 phr of oak bark showed a slight increase in the degree of cross-linking, which was an exception compared with the rest of the specimens.

### 3.4. The Influence of Oak Bark on the Mechanical Properties of NR Vulcanizates

Tensile testing is a method that evaluates rubber products for their applications. The incorporation of a filler changes the mechanical properties of composites. By selecting the correct filler–matrix pair and factors such as particle shape, interaction with the matrix, amount of filler, and particle size, the material can gain new properties [74]. Table 5 shows the results of the tensile strength test.

Of the composites tested, the NR10S sample achieved the highest TS_b_ value (15.8 MPa), indicating that it was the most durable. The addition of oak bark caused a decrease in tensile strength relative to the reference sample (except NR10S). The reason for this phenomenon was the agglomeration of the filler and the generation of stresses within the material. Modification of oak bark with n-octadecyltrimethoxysilane led most rubber compositions to improve their mechanical strength compared with their counterparts. It is worth mentioning that in the case of samples with 10 phr oak bark (NR10 and NR10S), the addition of silane improved the tensile strength by 26% (from 12.5 to 15.8 Mpa). It was concluded that the modified bio-filler became more compatible with the matrix, which contributed to the increase in TS_b_. For all samples, it was observed that there was an optimal amount of oak bark in the composition for which the sample reached a maximum TS_b_ value. For samples without filler modification, this value oscillated around 15 phr, while for samples with silane, it was around 10 phr. Figure 7 shows the dependence of the tensile strength on the bark content in NR vulcanizates. For samples with silane, the maximum shifts towards smaller amounts of oak bark in the NR compounds. It was noted that none of the samples tested achieved TS_b_ values below 10 Mpa.

One-way ANOVA showed a significant difference in tensile strength between unmodified (F = 36.8; *p* < 0.001) and modified (F = 64.7; *p* < 0.001) specimens. Pairwise multiple comparisons with Tukey’s HSD test (a = 0.05) determined a ranking of different composition subsets with statistical significance. All probability values less than 0.05 mean that H_0_ is false, and the results are not equal and do not include statistical error. For samples without silane, filling the sample with oak bark did not significantly change, so the point on the graph for sample NR15 can be considered a statistical error because only the NR25 specimen differed. For compositions with silane, it should be noted that sample NR10S differed significantly from most vulcanizates filled with larger amounts of oak bark. In addition, it is worth mentioning that silane modification led to statistically significant differences in TS_b_ values for only the composition with 10 phr of oak bark (F = 21.0; *p* = 0.004). For the remaining samples, the modification did not implement a change in tensile strength that was statistically significant. For all samples, the critical value of F for Snedecor’s F-distribution with r-1 degrees of freedom and n-r degrees of freedom in the denominator was determined, where r is the number of populations and n is the number of tests. The critical value for F_(1.8)_ = 5.32. Table 6 shows the significant difference between modified and unmodified vulcanizates.

This study showed that the reference sample achieved the highest elongation (E_b_ = 1287%). Similarly to the tensile strength, filled specimens reached maximum E_b_ at a certain amount of filler. For samples without silane, it was approx. 15 phr, while for those with silane, it was about 5 phr. All filled samples recorded lower relative elongation values than NR0. For the filled compositions, the NR5S vulcanizate reached the highest value of E_b_ = 1242%, but each of the filled samples showed high values of relative elongation. The decrease in elongation for the filled samples was probably due to the formation of agglomerates, which caused the polymeric matrix to weaken in stress transmission and eventually caused the material to break at lower elongation.

Stresses at elongation allowed assessment of the material’s stiffness and resistance to deformation. The highest stresses at 100% elongation (1 Mpa) were achieved by the NR25 sample. All filled specimens reached higher values of stress at 100% elongation than the reference sample, which proved that the addition of oak bark stiffened the structure of the specimen and increased resistance to deformation. Figure 8 illustrates a silanization scheme using oak bark to improve interfacial interactions and obtain better dispersion in the elastomeric matrix and improved hydrophobicity of the bark.

One of the issues affecting rubber products is that they are sensitive to degradation under the influence of oxygen, ozone, dynamic stresses, and also heat [74,75,76]. Therefore, anti-aging substances are very often used, which create stable complexes or react with free radicals formed by oxidation. These substances are divided into phenolic or amine derivatives. The results in Table 5 show that the addition of oak bark increased the resistance to thermo-oxidative aging. The values of the aging factor increased by at least twice in comparison with the NR0 sample (AF = 0.36); these results are satisfying, as they confirm the fact that the used filler can be considered as an anti-aging agent. The NR10 composition obtained an aging factor value higher than 1 (AF = 1.06) because this sample improved its mechanical properties after aging (the TS_b_ value increased). For the other vulcanizates, the AF value was in the range of 0.72–0.97. After thermo-oxidative aging, the referenced sample, despite its increased tensile strength, achieved relative elongation less than half the value and, due to the increase in S_e100_, was much stiffer than before aging. Such a phenomenon is unacceptable from the perspective of rubber products, which are exposed to constant stress in conditions where thermo-oxidative aging can occur. Therefore, it is important to incorporate a proper filler to provide anti-aging properties to the rubber. For filled samples, the decrease in E_b_ value was not as significant as for the NR10 composite (E_b_ = 1196%; E_b_’ = 1133%). In addition, the change in TS_b_ value after aging did not cause a major decrease in mechanical strength; for example, for the NR15 sample, the value dropped from 13.5 to 12.4 MPa. The addition of silane in most cases (except the sample with 10 phr of oak bark) resulted in an increase in resistance to thermo-oxidative aging (from 0.74 to 0.97 for compositions with 25 phr oak bark). The specimens with an oak bark content of more than 10 phr achieved higher resistance to aging than the others. Stress at elongation increased for each vulcanizate, due to stiffening and extra cross-linking under the influence of temperature. This phenomenon occurs sometimes with rubber products, but it is not advantageous due to the material becoming brittle and breaking at much lower elongation, as was the case for sample NR0. The highest stresses at elongation were obtained for the composite with the highest bark content, namely NR25 (S_e100_’ = 1.82 Mpa). Specimens NR10, NR15S, NR20S, and NR25S each recorded an increase in TS_b_ and a decrease in E_b_ values. Figure 9 shows the value of the aging factor for all filled samples and the relation between resistance to thermo-oxidation aging and the content of oak bark in the composite.

Investigating hysteresis losses enables the quantification of mechanical energy dissipated during sample deformation, which is not retained in the material for restoring its original form upon stress removal. Instead, this energy is converted into thermal energy. The calculated Mullins effect is associated with stress reduction observed during the next deformation of filled vulcanizates. This phenomenon is due to the breakdown of interactions between the filler and rubber, as well as the disintegration of filler aggregates within the matrix [66,67]. Table 7 summarizes the results of the hysteresis test. The lowest ΔW_1_ and ΔW_5_ values were achieved by the NR5 vulcanizate (34.77 N·mm and 24.23 N·mm, respectively) and NR5S vulcanizate (35.32 N·mm and 24.08 N·mm, respectively), while the highest losses were recorded for the NR25 sample (ΔW_1_: 67.97 N·mm, ΔW_5_: 32.94 N·mm). The addition of silane caused an increase in hysteresis losses in most of the compositions, except for the NR25S sample, where it was associated with a smaller loss than the NR25 compound. The calculated Mullins effect assumed the lowest values for the NR5 and NR10S vulcanizates (16.1% and 17.8%, respectively), proving that these compositions had the least filling agglomerates and the best damping properties among the tested composites. The highest E_M_ values were obtained by samples with the largest amount of bio-filler (NR25, NR25S), which confirms the presence of agglomerates in the structure of the sample. A trend was observed that a higher content of oak bark caused an increase in E_M_ and worse damping properties. Figure 10 shows the hysteresis loops for three different samples: unfilled vulcanizate (NR0), vulcanizate filled with oak bark (NR15), and vulcanizate filled with modified oak bark (NR15S). The selection of these compositions enabled the assessment of the filler’s influence on hysteresis losses and the behavior of different sample types during testing. It was noted that the reference sample showed the least hysteresis loss. Moreover, it is worth mentioning that the incorporation of oak bark did not reduce the hysteresis loss. The loops for the NR15 and NR15S compositions shifted significantly relative to the first stretching cycle, while for the NR0 sample, all loops overlapped.

The tear resistance is summarized in Table 7, and it was noted that the addition of bark caused a decrease in tear resistance, as each filled composition recorded lower T_s_ values than the reference. For the filled specimens, the NR10 composite achieved the highest tear resistance (T_s_ = 6.20/mm). It was observed that like tensile strength, the optimal amount of bark content is required to provide the best tear resistance. Based on the analysis, samples without the addition of silane required a bark content of 10 phr, whereas the composition without silane required about 15 phr of bark (for NR15S; T_s_ = 5.79 N/mm).

The hardness of a material is determined by its elastic properties, specifically Young’s modulus. This property allows us to determine changes in hardness from the surface to the depth of the material. In the case of rubber composites, the hardness is dependent on the amount of filler incorporated into the elastomeric matrix [67]. The results of the hardness tests are summarized in Table 6, indicating that the highest hardness levels were achieved by the NR25 and NR25S compositions, which recorded 57.6 °ShA and 59.6 °ShA, respectively. Each filled sample was harder than the reference specimen. It is important to mention that by modifying oak bark with n-octadecyltrimethoxysilane and adding it to the vulcanizate, the hardness level of the material was increased by an average of 2.6%. The research also confirms that all compositions containing silane showed higher hardness levels compared with those without it. This is due to the smaller agglomerates and better dispersion of the filler in the elastomeric matrix (compare Figure 3 and Figure 4), resulting in better polymer–filler interactions that give the material new properties. Figure 11 shows a linear trend between the amount of oak bark in the samples and the Shore A hardness. It was also observed that the incorporation of oak bark caused an increase in the degree of cross-linking, which had a direct impact on the greater hardness of the tested compositions.

The results of the analysis of variance for the compositions tested showed that the samples differed in unmodified F = 16.9; *p* < 0.001 and modified F = 44.7; *p* < 0.001. Tukey’s test allowed us to calculate results that were significantly different from each other. For this analysis, the Snedecor F-distribution with r-1 degrees of freedom and n-r degrees of freedom in the denominator was also used. Statistically significant changes in the hardness results after modification of filler were recorded for NR20S and NR25S. The results are summarized in Table 8.

### 3.5. The Influence of Oak Bark on the Hydrophobicity of NR Vulcanizates

The hydrophobicity of a surface can be determined by measuring its contact angle, as shown in the figures below. A surface is considered hydrophobic if it has a contact angle of over 90° [77]. However, when a lignocellulosic material like oak bark is added, the hydrophobicity of the composite decreases. To increase the hydrophobicity of such compositions (where the matrix is hydrophobic and the filler is hydrophilic), a substance can be applied to increase the compatibility of the filler with the polymeric matrix [78]. In this research, n-octadecylotrimethoxysilane was used to modify the hydroxyl groups on the surface of the oak bark and replace them with long aliphatic chains, which increased the hydrophobicity. The study’s results are presented below, and Figure 12 shows a droplet on the surfaces of selected NR samples.

In this study, the compositions with added hydrophilic fillers were found to be hydrophobic, with all samples having a contact angle exceeding 90°. The results of the contact angle test for the filled compositions are illustrated in Figure 13, indicating that the addition of increasing amounts of oak bark caused a gradual shift in the contact angle towards hydrophilicity. This is because oak bark is naturally created to retain moisture, despite changing weather conditions [79,80]. According to Ilek [78], the hygroscopicity of the bark depends on the species of tree from which it derives. For example, oak bark was able to store about 15% of water on average in this investigation. Therefore, the addition of an unmodified bio-filler such as oak bark increased the hydrophilicity of the NR biocomposite. To remove excess absorbed water, the filler was dried before being incorporated into the elastomeric matrix, to make the composite as hydrophobic as possible. When smaller amounts of oak bark were incorporated, no significant changes in the contact angle were observed. However, the addition of 10 phr of filler reduced the contact angle by about 10°. Nonetheless, for the NR15, NR20, and NR25 compositions, the contact angle values oscillated between 101° and 105° without any significant change in surface character. Oak bark modified with n-octadecylotrimethoxysilane had a positive effect on the hydrophobicity of the tested vulcanizates. All samples with the addition of silane (except NR5S) had a greater contact angle value than compositions with the same amount of filler. The NR25S vulcanizate achieved the highest contact angle value. The contact angle of samples containing 25 phr bark content was increased by up to 25% through the modification with oak bark.

The results suggest that adding silane to the NR compositions filled with oak bark significantly improved their hydrophobicity. The incorporation of n-octadecylotrimethoxysilane enhanced the compatibility between the elastomeric matrix and the filler, which resulted in higher contact angle values. Overall, this study concludes that the modification of oak bark with n-octadecylotrimethoxysilane can be an effective approach to enhance the hydrophobicity of vulcanizates.

A one-way analysis of variance (ANOVA) confirmed the heterogeneity of the results. Tukey’s test was then performed to find statistically different samples, which were then matched, as shown in Table 9. In addition, an analysis was carried out to determine whether the modification affected the contact angle results in any way. The results are summarized in Table 10 and confirm the differences between the tested vulcanizates. Snedecor’s F-distribution with r-1 degrees of freedom and n-r degrees of freedom in the denominators was used; due to the different numbers of trials (n), the critical value of F varied.

## 4. Conclusions

Oak bark can be used as a bio-filler in elastomeric matrices such as natural rubber. The above research shows that vulcanizate can achieve different properties depending on the content of this bio-filler. Among the filled samples, it was the NR10 composition that showed the shortest cross-linking time. The NR10S sample showed the best mechanical properties. It is worth noting that there is an optimum quantity of filler addition from which the sample obtains the best mechanical properties. For compositions with silane, the optimal content of oak bark was 10 phr but for the unmodified samples, the optimal content of oak bark was 15 phr.

Oak bark, known for its phenolic derivatives, serves not only as a bio-filler but also as an anti-aging agent in materials. Its eco-friendly nature reduces environmental pollution and may reduce the carbon footprint of products compared with traditional fillers like silica or carbon black. Specimens filled with oak bark significantly had enhanced anti-aging properties, with each sample achieving at least double the aging factor of the reference sample. Since natural rubber is not resistant to thermo-oxidative aging, adding oak bark is a good alternative [81,82,83,84,85].

The use of oak bark in NR vulcanizates led to an impact on their hydrophobicity. The NR25S composite showed the highest level of hydrophobicity, but other vulcanizates also exhibited this characteristic. However, the addition of oak bark to the NR matrix resulted in a reduction of the contact angle value and a decrease in the hydrophobicity of the material. This is due to the chemical composition of the bark and the presence of hydroxyl groups on its surface. On the other hand, when oak bark was modified with n-octadecylotrimethoxsilane, the hydrophobicity of the material increased despite the presence of a hydrophilic filler. Research has proven that the hydrophobicity of produced materials increases with the content of modified oak bark. The modified oak bark allowed the composite to achieve enhanced hydrophobicity.

The study found that oak bark can substitute synthetic fillers and anti-aging agents in composites, though requiring a surface modifier for compatibility with elastomeric matrices. This modification yields composites with optimal mechanical properties and hydrophobicity. Biocomposites align with the Sustainable Development Goals [19,20,21,22,23] by innovatively utilizing natural materials, promoting sustainable and eco-friendly technologies, and utilizing renewable resources for production. Biocomposites offer advantages such as comparable properties to traditional composites, environmental friendliness, and reduced carbon footprint, thus supporting sustainability objectives [86,87].

## Figures and Tables

**Figure 1 materials-17-01968-f001:**
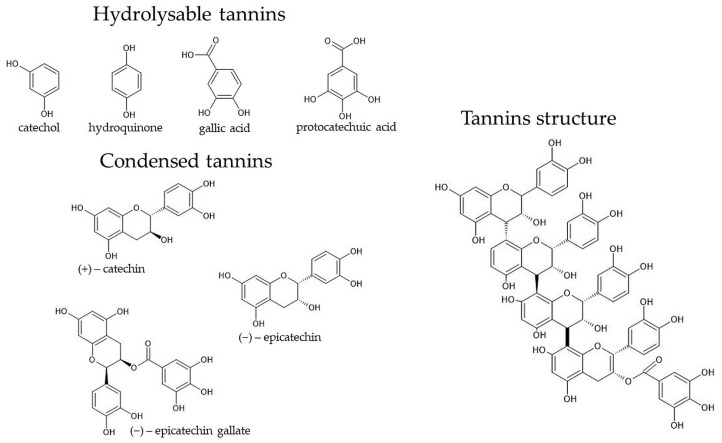
Types and structure of tannins contained in tree bark [58].

**Figure 2 materials-17-01968-f002:**
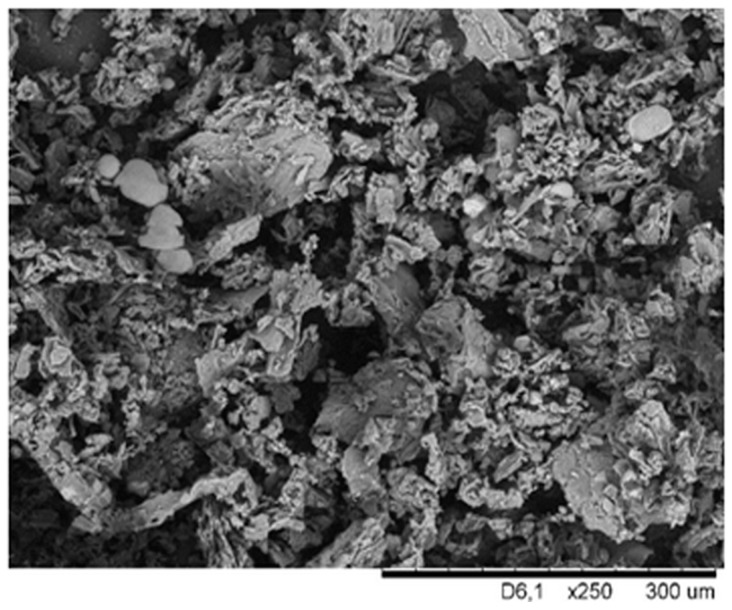
SEM photograph of oak bark at magnification 250×.

**Figure 3 materials-17-01968-f003:**
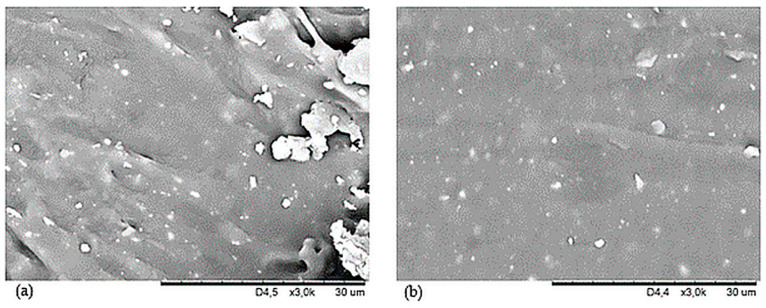
SEM photographs of the surface of vulcanizates: (**a**) NR15—3000×, (**b**) NR15S—3000×.

**Figure 4 materials-17-01968-f004:**
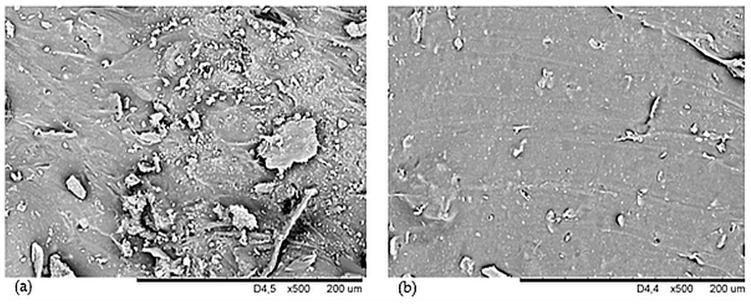
SEM photographs of the cross-section of samples: (**a**) NR15—500× and (**b**) NR15S—500×.

**Figure 5 materials-17-01968-f005:**
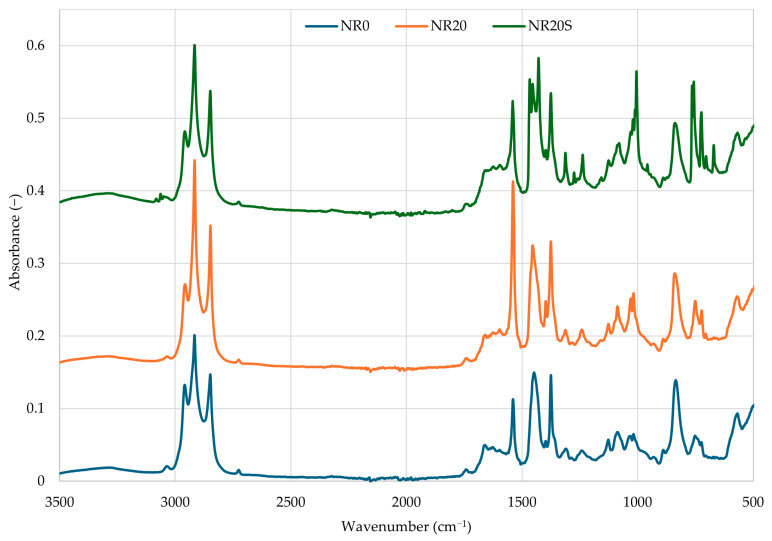
Infrared spectra of NR composites before (NR0) and after filling with unmodified oak bark (NR20) or N-OTMS-modified oak bark (NR20S).

**Figure 6 materials-17-01968-f006:**
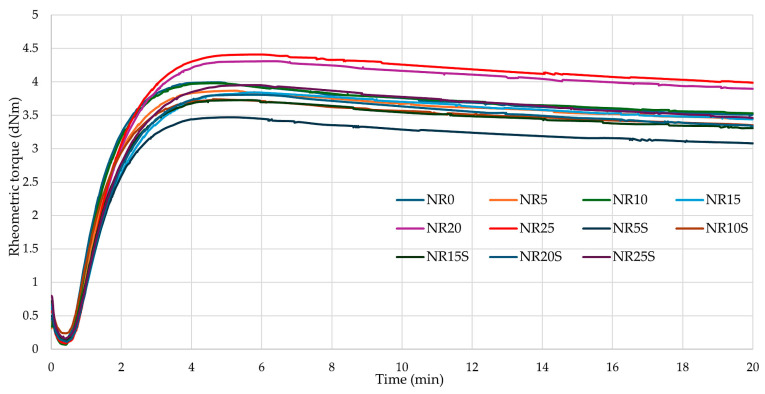
Vulcametric kinetics of NR composites filled with oak bark.

**Figure 7 materials-17-01968-f007:**
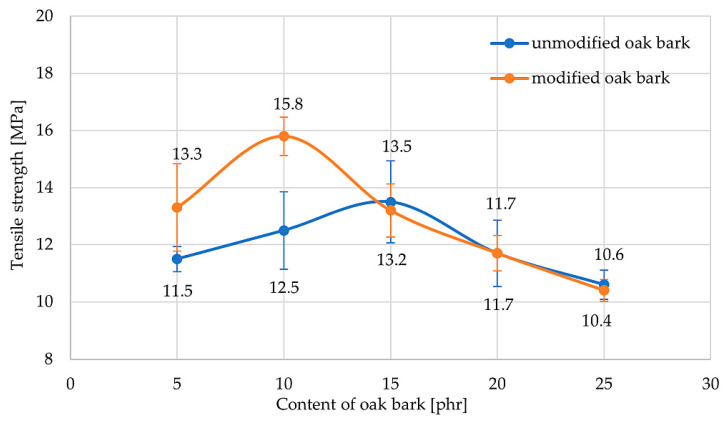
Influence of oak bark content on tensile strength of NR vulcanizates.

**Figure 8 materials-17-01968-f008:**
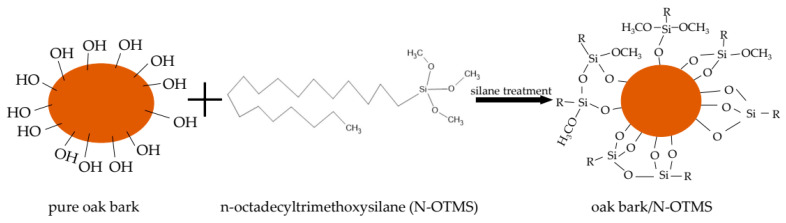
The reaction scheme of silane treatment (O—oxygen; C—carbon; Si—silicon; H—hydrogen; R—alkyl chain).

**Figure 9 materials-17-01968-f009:**
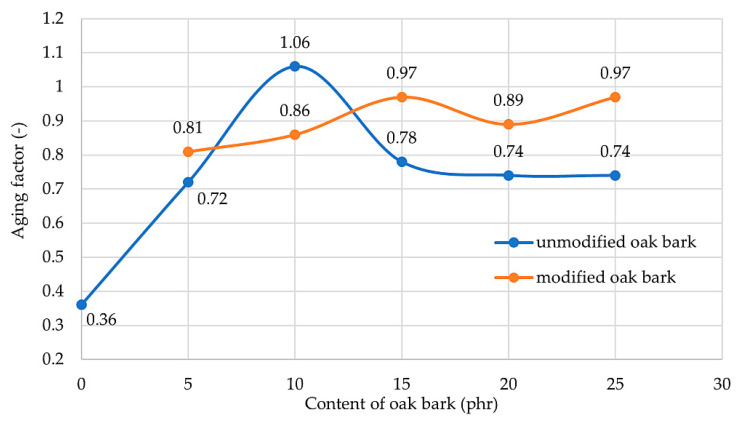
Resistance on thermo-oxidation aging for all filled samples.

**Figure 10 materials-17-01968-f010:**
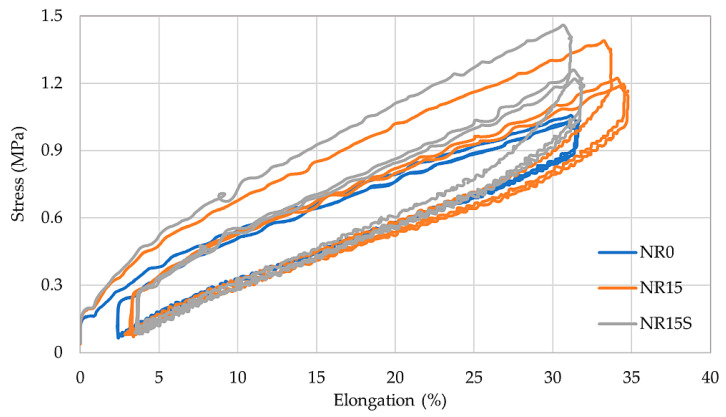
Hysteresis loops for unfilled sample (NR0), filled with oak bark sample (NR15), and filled with modified oak bark sample (NR15S).

**Figure 11 materials-17-01968-f011:**
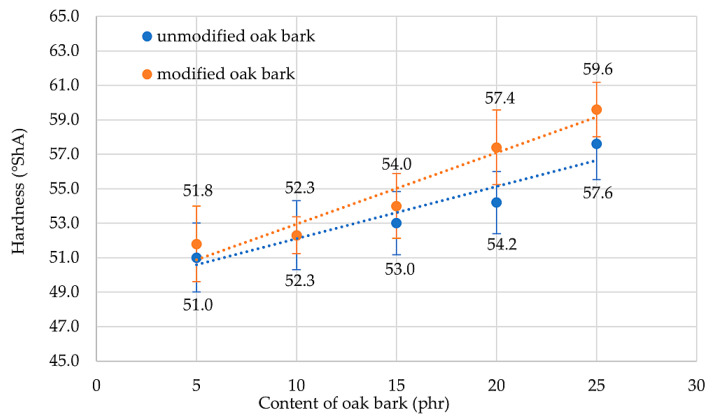
The hardness of NR vulcanizates containing oak bark.

**Figure 12 materials-17-01968-f012:**
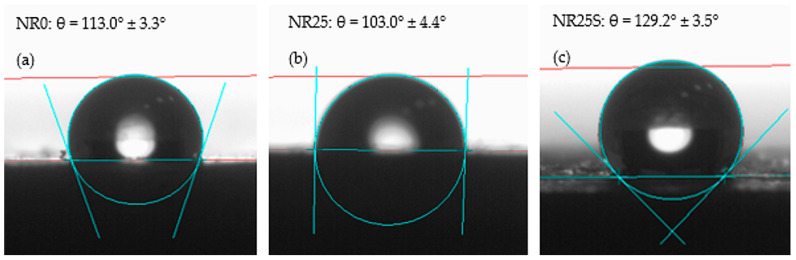
Pictures of a water droplet on the surface of (**a**) NR, (**b**) NR25, (**c**) NR25S.

**Figure 13 materials-17-01968-f013:**
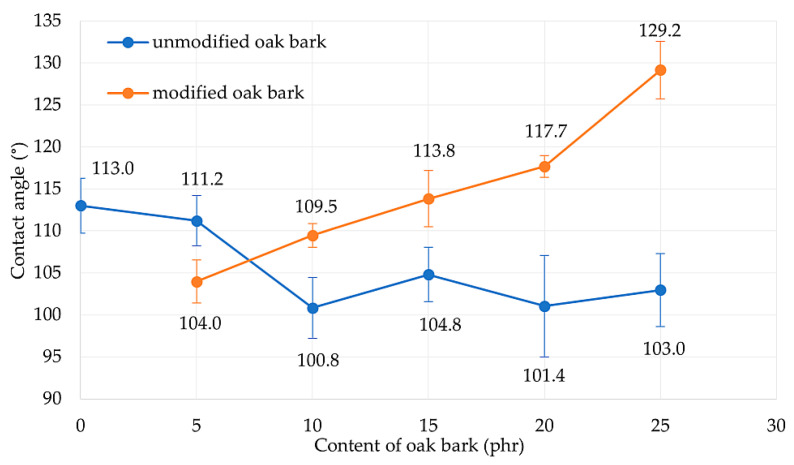
Contact angle measurements for NR/oak bark vulcanizates.

**Table 1 materials-17-01968-t001:** Compositions of the NR compounds.

Ingredient	Ingredient Amount (phr)										
NR	100	100	100	100	100	100	100	100	100	100	100
S_8_	2	2	2	2	2	2	2	2	2	2	2
ZnO	5	5	5	5	5	5	5	5	5	5	5
SA	2	2	2	2	2	2	2	2	2	2	2
MBT	1.5	1.5	1.5	1.5	1.5	1.5	1.5	1.5	1.5	1.5	1.5
OB	-	5	10	15	20	25	5	10	15	20	25
N-OTMS	-	-	-	-	-	-	0.5	1	1.5	2	2.5
Composite symbol	NR0	NR5	NR10	NR15	NR20	NR25	NR5S	NR10S	NR15S	NR20S	NR25S

NR—natural rubber, S_8_—sulfur, ZnO—zinc oxide, MBT—mercaptobenzothiazole, SA—stearic acid, OB—oak bark, N-OTMS—n-octadecyltrimethoxysilane, phr—parts per hundred of rubber.

**Table 2 materials-17-01968-t002:** Absorption bands characteristic NR composites before (NR0) and after filling with unmodified oak bark (NR20) or N-OTMS-modified oak bark (NR20S).

Wavenumber (cm^−1^)	Band Intensity	Chemical Group	Composite		
			NR	NR20	NR20S
2958	Medium	C-H in -CH_3_	+	+	+
2916	Intensive, sharp	C-H in -CH_3_ and -CH_2_-	+	+	+
2848	Medium	C-H in -CH_3_ and -CH_2_-	+	+	+
1680–1596	Weak, broad	>C=C<	+	+	+
1538	Medium, sharp	>C=C<	+	+	+
1454–1428	Medium doublet	Si-CH_3_	−	−	+
1444	Intensive, sharp	C-H in -CH_3_ and -CH_2_-	+	+	+
1375	Medium	C-H in -CH_3_	+	+	+
1275	Weak, sharp	Si-CH_2_	−	−	+
1242	Weak	C-C in CH_3_	+	+	+
1126	Weak	C-C in CH_3_	+	+	+
1080	Weak	C-H in -CH_2_-	+	+	+
1020–1005	Intensive, sharp	Si-O	−	−	+
840	Intensive	>C=C<	+	+	+
765	Intensive, sharp	C-Si-C	−	−	+
700–660	Medium doublet	Si in C-Si-C	−	−	+

+ means the presence of a band in the spectrum of the indicated vulcanizate, and − means the absence of a band in the spectrum of the indicated vulcanizate.

**Table 3 materials-17-01968-t003:** Vulcametric parameters of NR composites filled with oak bark.

NR Sample	Properties						
	T_min_ (dNm)	T_max_ (dNm)	T_20_ (dNm)	∆T_R_ (dNm)	t_02_ (min)	t_90_ (min)	CRI (min^−1^)
NR0	0.16	3.99	3.35	0.64	1.30	2.49	84.03
NR5	0.11	3.87	3.34	0.53	1.40	2.74	74.63
NR10	0.07	3.98	3.46	0.52	1.30	2.55	80.00
NR15	0.13	3.84	3.31	0.53	1.62	3.21	62.89
NR20	0.11	4.31	3.79	0.52	1.47	3.03	64.10
NR25	0.09	4.41	3.90	0.51	1.44	3.04	62.50
NR5S	0.14	3.47	2.94	0.53	1.62	2.78	86.21
NR10S	0.24	3.74	3.11	0.63	1.44	2.68	80.64
NR15S	0.13	3.73	3.18	0.55	1.52	2.84	75.76
NR20S	0.11	3.81	3.23	0.58	1.52	2.98	68.49
NR25S	0.16	3.95	3.30	0.65	1.55	3.09	64.93

T_min_—minimal rheometric torque; T_max_—maximal rheometric torque; T_20_—rheometric torque after 20 min of heating; ∆T_R_—reversion losses of rheometric torque; t_02_—scorch time; t_90_—optimal vulcanization time; CRI—cure rate index.

**Table 4 materials-17-01968-t004:** Cross-linking parameters determined by equilibrium swelling of NR vulcanizates filled with oak bark.

NR Sample	Properties			
	Q_v_ (mL/mL)	−Q_w_ (mg/mg)	V_R_ (-)	α_c_ (-)
NR0	5.47 ± 0.08	0.06 ± 0.01	0.155 ± 0.01	0.183 ± 0.01
NR5	5.24 ± 0.38	0.06 ± 0.01	0.161 ± 0.01	0.191 ± 0.01
NR10	5.21 ± 0.07	0.06 ± 0.01	0.161 ± 0.01	0.192 ± 0.01
NR15	5.36 ± 2.20	0.02 ± 0.32	0.172 ± 0.06	0.187 ± 0.09
NR20	4.48 ± 1.14	0.06 ± 0.01	0.190 ± 0.05	0.223 ± 0.08
NR25	4.28 ± 0.06	0.06 ± 0.01	0.189 ± 0.01	0.234 ± 0.01
NR5S	5.53 ± 0.06	0.06 ± 0.01	0.153 ± 0.01	0.181 ± 0.01
NR10S	5.11 ± 0.08	0.06 ± 0.01	0.164 ± 0.01	0.196 ± 0.01
NR15S	5.21 ± 0.44	0.07 ± 0.01	0.162 ± 0.01	0.192 ± 0.02
NR20S	4.84 ± 0.11	0.07 ± 0.01	0.171 ± 0.01	0.207 ± 0.01
NR25S	5.23 ± 0.15	0.07 ± 0.01	0.161 ± 0.01	0.191 ± 0.01

Q_v_—equilibrium volume swelling in toluene; −Q_w_—content of the eluted fraction in toluene; V_R_—volume fraction of rubber in swollen material in toluene; α_c_—degree of cross-linking.

**Table 5 materials-17-01968-t005:** Strength properties of NR vulcanizates filled with oak bark before and after the thermo-oxidative aging process.

NR Sample	Properties						
	S_e100_ (MPa)	TS_b_ (MPa)	E_b_ (%)	S_e100_* (MPa)	TS_b_* (MPa)	E_b_* (%)	AF (-)
NR0	0.64 ± 0.01	14.9 ± 0.5 ^a,d^	1287 ± 1	0.87 ± 0.02	16.3 ± 1.3	424 ± 156	0.36
NR5	0.65 ± 0.04	11.5 ± 1.4 ^b,c^	1096 ± 70	0.77 ± 0.01	9.8 ± 0.3	917 ± 26	0.72
NR10	0.73 ± 0.05	12.5 ± 1.5 ^a,b,c^	1196 ± 55	0.74 ± 0.03	14.0 ± 0.8	1133 ± 33	1.06
NR15	0.77 ± 0.02	13.5 ± 1.2 ^a,b^	1212 ± 63	0.86 ± 0.02	12.4 ± 0.4	1028 ± 41	0.78
NR20	0.81 ± 0.02	11.7 ± 0.6 ^b,c^	1136 ± 31	0.84 ± 0.08	10.2 ± 0.1	963 ± 19	0.74
NR25	1.00 ± 0.03	10.6 ± 0.4 ^c^	980 ± 24	1.14 ± 0.02	9.5 ± 0.7	811 ± 39	0.74
NR5S	0.63 ± 0.01	13.3 ± 1.5 ^d,e,f^	1242 ± 37	0.73 ± 0.01	12.6 ± 1.8	1064 ± 69	0.81
NR10S	0.72 ± 0.02	15.8 ± 0.7 ^d^	1226 ± 26	0.76 ± 0.04	14.7 ± 1.0	1133 ± 46	0.86
NR15S	0.73 ± 0.02	13.2 ± 0.9 ^d,e^	1182 ± 44	0.93 ± 0.02	14.5 ± 0.2	1043 ± 12	0.97
NR20S	0.81 ± 0.02	11.7 ± 0.6 ^e^	1095 ± 10	0.97 ± 0.04	11.9 ± 0.7	962 ± 19	0.89
NR25S	0.86 ± 0.03	10.4 ± 0.4 ^f^	1016 ± 35	1.05 ± 0.06	11.0 ± 0.9	927 ± 41	0.97

S_e100_—stress at an elongation of 100%; TS_b_—tensile strength; E_b_—elongation at break; S*_e100_—stress at an elongation of 100% after thermo-oxidative aging; TS_b_*—tensile strength after thermo-oxidative aging; E_b_*—elongation at break after thermo-oxidative aging; AF—aging factor; superscript letters indicate statistically homogeneous subsets (Tukey’s HSD test, α = 0.05).

**Table 6 materials-17-01968-t006:** The significant difference between modified and unmodified vulcanizates for tensile strength (α = 0.05, r = 2, n = 10).

Parameters	Content of Filler				
	5	10	15	20	25
F_(1.8)_	5.32	5.32	5.32	5.32	5.32
F value	3.95	21.0	0.234	0.027	0.586
*p*	0.082	0.004	0.641	0.873	0.485
Significant difference	NO	YES	NO	NO	NO

**Table 7 materials-17-01968-t007:** Tear resistance, hardness, hysteresis losses, and Mullins effect of NR vulcanizates filled with oak bark.

NR Sample	Properties				
	T_s_ (N/mm)	HA (°ShA)	ΔW_1_ (N·mm)	ΔW_5_ (N·mm)	E_M_ (%)
NR0	7.18 ± 0.73	48.3 ± 2.2 ^a^	30.33	24.17	9.4
NR5	5.85 ± 0.01	51.0 ± 2.0 ^a,b^	34.77	24.23	16.1
NR10	6.20 ± 0.33	52.3 ± 2.0 ^b,c^	43.44	28.65	18.9
NR15	4.28 ± 0.03	53.0 ± 1.9 ^b,c^	50.13	31.03	23.9
NR20	4.84 ± 0.07	54.2 ± 1.8 ^c^	53.75	27.99	37.4
NR25	4.73 ± 0.15	57.6 ± 2.1 ^d^	67.97	32.94	38.9
NR5S	4.83 ± 0.31	51.8 ± 2.2 ^e^	35.32	24.08	18.8
NR10S	5.18 ± 0.75	52.3 ± 1.1 ^e^	43.65	29.02	17.8
NR15S	5.79 ± 1.52	54.0 ± 1.9 ^e^	49.00	26.80	31.0
NR20S	5.57 ± 1.03	57.4 ± 2.2 ^f^	56.83	29.26	35.7
NR25S	4.53 ± 0.33	59.6 ± 1.6 ^g^	64.36	32.14	38.7

T_s_—tear resistance; HA—Shore A hardness; ΔW_1_. ΔW_5_—hysteresis losses during the first and fifth sample stretching cycles; E_M_—Mullins effect; superscript letters indicate statistically homogeneous subsets (Tukey’s HSD test, α = 0.05).

**Table 8 materials-17-01968-t008:** The significant differences between modified and unmodified vulcanizates for hardness (α = 0.05, r = 2, n = 20).

Parameters	Content of Filler				
	5	10	15	20	25
F_(1.18)_	4.414	4.414	4.414	4.414	4.414
F value	1.770	0.078	1.050	7.620	5.630
*p*	0.200	0.784	0.320	0.014	0.032
Significant difference	NO	NO	NO	YES	YES

**Table 9 materials-17-01968-t009:** Water contact angle—ANOVA analysis and Tukey’s test for tested NR vulcanizates (α = 0.05).

Unmodified (F = 16.3; *p* < 0.001)						
Sample	Parameters	NR5	NR10	NR15	NR20	NR25
NR0	t value	1.59	12.70 *	8.20 *	11.95 *	10.03 *
	*p*	0.973	<0.001	0.001	<0.001	<0.001
NR5	t value	-	11.20 *	6.61 *	10.36 *	8.44 *
	*p*	-	<0.001	0.007	<0.001	0.003
NR10	t value	-	-	4.55	0.80	2.71
	*p*	-	-	0.082	0.999	0.749
NR15	t value	-	-	-	3.75	1.84
	*p*	-	-	-	0.354	0.920
NR20	t value	-	-	-	-	1.92
	*p*	-	-	-	-	0.950
**Modified (F = 59.6; *p* < 0.001)**						
**Sample**	**Parameters**	**NR5**	**NR10**	**NR15**	**NR20**	**NR25**
NR0	t value	9.05 *	3.54	0.83	4.67	16.20 *
	*p*	<0.001	0.335	0.993	0.099	<0.001
NR5S	t value	-	5.51 *	9.87 *	13.72 *	25.20 *
	*p*	-	0.020	<0.001	<0.001	<.001
NR10S	t value	-	-	4.364	8.21 *	19.70 *
	*p*	-	-	0.091	<0.001	<0.001
NR15S	t value	-	-	-	3.85	15.30 *
	*p*	-	-	-	0.177	<0.001
NR20S	t value	-	-	-	-	11.50 *
	*p*	-	-	-	-	<0.001

* *p* < 0.05—comparison determined statistically significant.

**Table 10 materials-17-01968-t010:** The significant difference in water contact angle between modified and unmodified vulcanizates (α = 0.05).

Parameters	Content of Filler				
	5	10	15	20	25
F critical	4.60	4.75	4.35	5.12	4.84
F value	26.9	62.7	39.7	43.0	142.0
*p*	<0.001	<0.001	<0.001	<0.001	<0.001
Significant difference	YES	YES	YES	YES	YES

## Data Availability

Data are contained within the article.

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
