# Peer review of "Effect of Modified and Unmodified Oak Bark (Quercus Cortex) on the Cross-Linking Process and Mechanical, Anti-Aging, and Hydrophobic Properties of Biocomposites Produced from Natural Rubber (NR)"

_materials, 2024, doi:10.3390/ma17091968_

Round 1
Reviewer 1 Report
Comments and Suggestions for Authors
The manuscript describes a study on the use of oak bark as a filler for preparing natural rubber composites and the investigations on the influence on the physicochemical properties. Some results are collected and discussed, suggesting that the modified oak bark as filler performs better in the vulcanizates in terms of the properties such as hydrophobicity, mechanical strength, etc. Although the work is interesting, some drawbacks are also very obvious, which should draw the authors’ attention to making a careful revision. The manuscript thus could be acceptable for publication in this journal if the following concerns are addressed.
1. The text regarding the research background of the manuscript is too long and requires a long time to complete the reading through. Some descriptions are not necessary, for example, only the water droplet images occupy a whole page (Figure 11); the section of the conclusions is also tedious to reading because it is also too long and many unimportant descriptions are involved. This means that the text of the manuscript could be shortened and many materials could be placed in a supporting material.
2. Contrary to some redundant descriptions that existing in the text, the information in the abstract is inadequate. Some key data regarding the properties and performance of the material are missing, and only general descriptions are given. The key data should be incorporated into the abstract for readers to understand the importance of the work.
Comments on the Quality of English LanguageThe language is excelent.
Reviewer 2 Report
Comments and Suggestions for Authors
The article “Effect of modified and unmodified oak bark (Quercus cortex) 2 on the cross-linking process, and mechanical, anti-aging and 3 hydrophobic properties of biocomposites produced from natural rubber (NR)” deals with the inclusion of oak bark as a filler in natural rubber. The oak bark was also derivatized with n-octadecyltrimethoxysilane and compared with the unmodified bark. The filler was added in different proportions. The obtained materials were characterized. The article is interesting but it needs major revision.
1) Abstract: The acronym NR is named but you have to go back to the title to know what it is
2) The most important aspect I have concerns is that it seems that no statistical analysis was performed. It should be done in order to compare the samples.
3) In the Introduction and the conclusion the Sustainable Development Goals are mentioned. A reference should be included.
4) The introduction is too long. I suggest being more concise.
5) In Materials and Methods:
Materials and Methods:
a) It is redundant to mention the two-roll mil’s characteristics twice.
b) Specify the special program used for measuring contact angle.
6) Line 372: Change "It was important to noted" to "It was important to note."
7) In the case of environmental friendliness, it seems unusual because the fillers that you mention to be replaced (carbon black or silica) may not be as toxic as you mention. Given the reagents used in the synthesis, it appears inaccurate to classify this material as environmentally friendly. Additionally, the reagent used to derivatize the filler is not environmentally friendly. I suggest avoiding these definitions.
8) Line 611: The paragraph is a bit mixed up; it discusses anti-aging and eco-friendly aspects together.
9) There is no assay to confirm the success of the derivatization. Consider using FTIR for this purpose.
Comments on the Quality of English LanguageThe english is correct.
Reviewer 3 Report
Comments and Suggestions for Authors
In this manuscript the authors report on the effect of oak bark with and without chemical modification. The effect of modified fillers was to improve all of the ageing resistance, mechanical properties, and hydrophobicity. The results are discussed in terms of structure-property relationships. The whole manuscript is well structured. I think this study is interesting enough to warrant publication. Some minor revisions are as follows.
1) The text, especially Introduction and Conclusion, is redundant.
2) It is recommended to check the chemical modification with a silane coupling by IR spectroscopy.
Reviewer 4 Report
Comments and Suggestions for Authors
Dear Editor and Authors,
Below you will find the comments and suggestions made about the manuscript materials-2930963 entitled “Effect of modified and unmodified oak bark (Quercus cortex) on the cross-linking process, and mechanical, anti-aging and hydrophobic properties of biocomposites produced from natural rubber (NR)” by Smejda-Krzewicka et al.
General comment:
This manuscript presents the development of elastomer-based composite materials, based on natural rubber (NR) reinforced with oak bark (Quercus cortex). The objective is to contribute to a new generation of sustainable composite materials using mostly plant-based additives as the reinforcing agent and the NR matrix itself. The manuscript is generally well structured, presents interesting and relevant results, but requires major revisions before being accepted for publication in Materials.
Specific comments:
- A thorough review of English grammar is suggested. Some sentences are long and confusing and reading fluency could be improved with appropriate punctuation.
- The abstract would benefit from including specific data on some of the improvements achieved. For example, quantify the improvement in aging resistance due to the incorporation of oak bark, or the decrease in hydrophobicity. This would help the reader place themselves in the context of the work.
- The authors mention that oak bark is presented as a “cost-effective” strategy but no data is discussed in this regard. Regarding what is cost-effective? Conventional fillers such as carbon black or silica? Quantify.
- According to the discussed literature, the authors seem to have overlooked recent literature published in high-impact journals, related to the development of bio-based NR compounds. This could be commented on, to give a more updated perspective (1-3 years) in the introduction. Some suggested references are: 1. Polymer Composites, 2024, DOI: 10.1002/pc.28313; 2. Polymer Composites, 2024, 45(5), 4524-4537, DOI: 10.1002/pc.28078. However, authors are encouraged to expand this list with a particular search.
- Some details to be specified in the methodology:
a. How did the mixing occur between the filler and the silane?
b. Why was a constant curing time of 4 min used for all samples and not the respective t90 of each formulation?
c. Is the surface area observed by SEM obtained by a cryogenic fracture?
d. For the tear tests, the accuracy of the 40 mm cut made (I assume a notch), how was it controlled? Or was it done by hand?
e. At what time is the hardness reading taken? Instant or after a waiting period?
- For the discussion of reversion in the curing curves, it would be convenient to present the curves in the main text or as Supporting Information.
- The larger the amount of bark, the longer the curing process. Explain why? Writing should not be limited to a simple report of results. The entire manuscript in general would be improved by contrasting the results obtained with the available literature.
- Considering the error bars, it seems that the only value where there is an improvement in mechanical resistance (TS) due to the incorporation of silane is with 10 phr of the filler. What is the reason for such specific value? Have similar results been found in other publications? For simple statistics, 5 different samples are being evaluated where 4 do not show relevant variation. Could 10 phr be a statistical error? Otherwise explain this change scientifically.
- The authors state the following: “After conducting a comparison of the effects of various bio-additives on natural rubber, it can be determined that adding oak bark has a positive impact on the mechanical properties of biocomposites. The use of oak bark as a filler leads to higher tensile strength than when starch, walnut shells, or eggshells are incorporated into an elastomeric matrix, while still maintaining higher elongations.” This comparison would ONLY be valid using the same recipe ingredients, same curing methodology (temperature, pressure), same NR grade. Otherwise, BE VERY CAREFUL. Add a warning or directly rethink the content of the sentence.
- The conclusion ends with the following sentence: “Additionally, using bark as a bio-filler helps prevent it from being wasted and allows it to be landfilled. As an alternative to traditional plastics, they can result in reduced pollution in the seas and reduce deforestation using waste products.” I strongly disagree with this statement. In order to talk about real sustainability, a life cycle analysis of the products that can be develop with this material is necessary. It is true that promoting the use of sustainable additives is positive, but it is not a guarantee that a material that is covalently crosslinked with sulfur and contains ZnO, can be landfilled “safely”. A material, even if it is NR reinforced with a natural fiber, crosslinked in this way, is not a sustainable one. Non-dynamic crosslinking is irreversible and unless recovered via alternative recycling (such as shredding), the material is not sustainable.
Comments on the Quality of English LanguageExtensive editing of English language required.
Round 2
Reviewer 2 Report
Comments and Suggestions for Authors
The corrections made by the authors fulfilled all my requirements
Reviewer 4 Report
Comments and Suggestions for Authors
The authors have adequately addressed the comments and suggestions made, which is why I recommend the publication of this manuscript in Materials.